SciPost Physics

# Hopf Exceptional Points

Tsuneya Yoshida[1,2*], Emil J. Bergholtz[3†] and Tomáš Bzdušek[4‡]

**1** Department of Physics, Kyoto University, Kyoto 606-8502, Japan
**2** Institute for Theoretical Physics, ETH Zurich, 8093 Zurich, Switzerland
**3** Department of Physics, Stockholm University,
AlbaNova University Center, 10691 Stockholm, Sweden
**4** Department of Physics, University of Zürich,
Winterthurerstrasse 190, 8057 Zürich, Switzerland

⋆ yoshida.tsuneya.2z@kyoto-u.ac.jp , † emil.bergholtz@fysik.su.se , ‡ tomas.bzdusek@uzh.ch

## Abstract

Exceptional points at which eigenvalues and eigenvectors of non-Hermitian matrices coalesce are ubiquitous in the description of a wide range of platforms from photonic or mechanical metamaterials to open quantum systems. Here, we introduce a class of *Hopf exceptional points* (HEPs) that are protected by the Hopf invariants (including the higher-dimensional generalizations) and which exhibit phenomenology sharply distinct from conventional exceptional points. Saliently, owing to their $\mathbb{Z}_2$ topological invariant related to the Witten anomaly, three-fold HEPs and symmetry-protected five-fold HEPs act as their own "antiparticles". Furthermore, based on higher homotopy groups of spheres, we predict the existence of multifold HEPs and symmetry-protected HEPs with non-Hermitian topology captured by a range of finite groups (such as $\mathbb{Z}_3$, $\mathbb{Z}_{12}$, or $\mathbb{Z}_{24}$) beyond the periodic table of Bernard-LeClair symmetry classes.

## 1  Introduction

The notion of topology plays a pivotal role in modern condensed matter physics of both quantum [1–5] and classical systems [6–10]. Saliently, topological physics implies the possibility of various exotic quasiparticles. One of the prime examples is a Majorana zero mode [11–14] whose antiparticle is itself. In addition, a Weyl fermion [15–21], protected by Chern number in topological semimetals, is a source of negative magnetoresistance [22] which is a signal of chiral anomaly. The periodic table for the ten Altland-Zirnbauer symmetry classes provides a systematic understanding of the topological obstructions inducing these exotic excitations in Hermitian systems [23–29].

Notably, open systems coupled to environments host topological excitations for which non-Hermiticity is essential, such as exceptional points [30–35]. At exceptional points, two energy bands touch in both the real and the imaginary parts. Such band touchings are protected by the winding topology of energy eigenvalues [36]. In sharp contrast to Hermitian topological excitations, exceptional points exhibit a dispersion with a fractional exponent. Exceptional points and their variants [36–50] are reported for a wide range of platforms from quantum systems [51–60] to metamaterials [61–74], indicating the ubiquity of these non-Hermitian excitations. In particular, the high controllability of synthetic systems allows for the realization of exceptional points in dimensions $d$ larger than three [61, 62, 75–77].

Among the various exceptional points, multifold exceptional points exhibit an $n$-fold band touching [76–84]. These unique excitations are beyond the existing classification table for Bernard-LeClair symmetry classes [85–89] and discussed in interdisciplinary fields [66, 75, 90–103]. However, topology of the formerly reported $n$-fold exceptional points (EP$n$'s) is generally characterized by $\mathbb{Z}$ invariants [77, 78]. This, in particular, implies that an exceptional point cannot be its own "antiparticle".

In this work, we report novel non-Hermitian topological excitations, dubbed $n$-fold Hopf exceptional points (HEP$n$'s, $n = 3, 4, 5$), which are topologically stabilized by higher homotopy groups of spheres, labeled $\pi_j(S^d)$, with integers $j > d \geq 2$. Saliently, an HEP3 and a symmetry-protected HEP5 exhibit unusual $\mathbb{Z}_2$ topology, meaning that they act as their own antiparticle. We trace this striking feature to the homotopy group $\pi_4(S^3) = \mathbb{Z}_2$ (see Fig. 1). We further discover symmetry-protected HEP4 whose topology is classified by $\pi_3(S^2) = \mathbb{Z}$. By systematically leveraging higher homotopy groups of spheres, we elucidate the potential presence of HEP$n$'s characterized by abundant finite groups (e.g., $\mathbb{Z}_3$, $\mathbb{Z}_{12}$, and $\mathbb{Z}_{24}$) beyond the existing classification table.

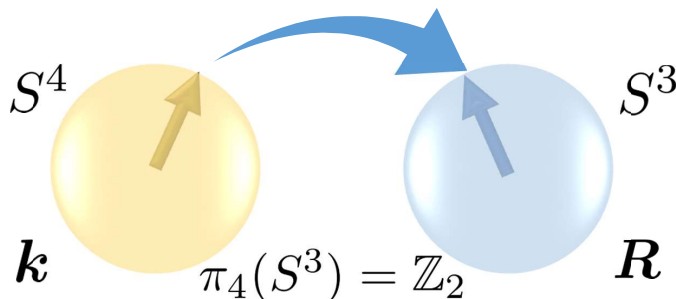

Figure 1: Illustration of $\mathbb{Z}_2$ topology protecting a three-fold Hopf exceptional point (HEP3). The resultant vector $\boldsymbol{R}(\boldsymbol{k})$ defines a map from a 4-sphere in the momentum (or parameter) space to a 3-sphere in the space of the resultant vector [see Eqs. (1) and (2)]. If the map is topologically nontrivial, an HEP3 with $\mathbb{Z}_2$ topology exists inside the 4-sphere.

## 2 HEP3s with $\mathbb{Z}_2$ topology

As a first example, we consider a generic three-band non-Hermitian Hamiltonian $H(\boldsymbol{k})$ in a five-dimensional momentum (or parameter) space denoted by $\boldsymbol{k} = (k_1, \ldots, k_5)$. Such a momentum space whose dimensionality is higher than three can be accessed in metamaterials [61,62,75–77]. The formation of an EP3 is captured by vanishing resultants $r_j \in \mathbb{C}$

$$r_j(\boldsymbol{k}) = \mathrm{Res}[\partial_E^{n-1-j} P(E, \boldsymbol{k}), \partial_E^{n-1} P(E, \boldsymbol{k})], \tag{1}$$

with $j = 1, 2, \ldots, n-1$ and $n = 3$ (see Appendix A.1). Here, $P(E, \boldsymbol{k}) = \det[H(\boldsymbol{k}) - E\mathbb{1}] \in \mathbb{C}$ is the characteristic polynomial, and $\partial_E$ denotes derivative with respect to the polynomial variable $E \in \mathbb{C}$.

To expose whether the EP3 is in fact an HEP3, we consider a 4-sphere $S^4 \subset \mathbb{R}^5$ enclosing the band touching. Assuming $(r_1, r_2) \neq (0, 0)$ on the 4-sphere, we introduce a normalized vector $\boldsymbol{n} = \boldsymbol{R}/\|\boldsymbol{R}\|$ ($\|\boldsymbol{R}\| = \sqrt{\boldsymbol{R} \cdot \boldsymbol{R}}$) with the resultant vector

$$\boldsymbol{R}(\boldsymbol{k}) = (\mathrm{Re}[r_1], \mathrm{Im}[r_1], \mathrm{Re}[r_2], \mathrm{Im}[r_2]). \tag{2}$$

Vector $\boldsymbol{n}(\boldsymbol{k})$ defines a map from $S^4$ to $S^3$ whose topology is classified by an element of the homotopy group $\pi_4(S^3) = \mathbb{Z}_2$ (see also Appendix A.2). When the map of $\boldsymbol{n}(\boldsymbol{k})$ possesses nontrivial $\mathbb{Z}_2$ topology, the enclosed band touching is an HEP3.

The $\mathbb{Z}_2$ invariant $\nu_F$ of maps from $S^4$ to $S^3$, originally discovered by Freudenthal [104], was considered in physics in the context of the Witten anomaly in SU(2) gauge theory [105] and to describe topological defects in superfluid ${}^3$He [106]. Numerical computation of this $\mathbb{Z}_2$ invariant is carried out by the following representation [107]

$$\nu_F = \frac{1}{4\pi} \oint d^4 p\, \epsilon^{\mu\nu\rho\lambda}[\partial_\mu \Delta\varphi(\boldsymbol{p})] A_\nu F_{\rho\lambda} \tag{3}$$

with $\mu, \nu, \rho, \lambda = 1, \ldots, 4$ and anti-symmetric tensor $\epsilon^{\mu\nu\rho\lambda}$ taking $\epsilon^{1234} = 1$. The Berry connection $A_\mu$ and the Berry curvature $F_{\mu\nu}$ are obtained from the *resultant Hamiltonian*, and the phase $\Delta\varphi(\boldsymbol{p})$ ($0 \leq \Delta\varphi \leq 2\pi$) is obtained from the resultant vector. For the precise definitions of $A_\mu$, $F_{\mu\nu}$ and $\Delta\varphi$, see Appendix A.3. Vector $\boldsymbol{p}$ parametrizes the 4-sphere in the momentum space. While the integral in Eq. (3) can take an arbitrary integer value, gauge transformations can change $\nu_F$ by multiples of two [105].

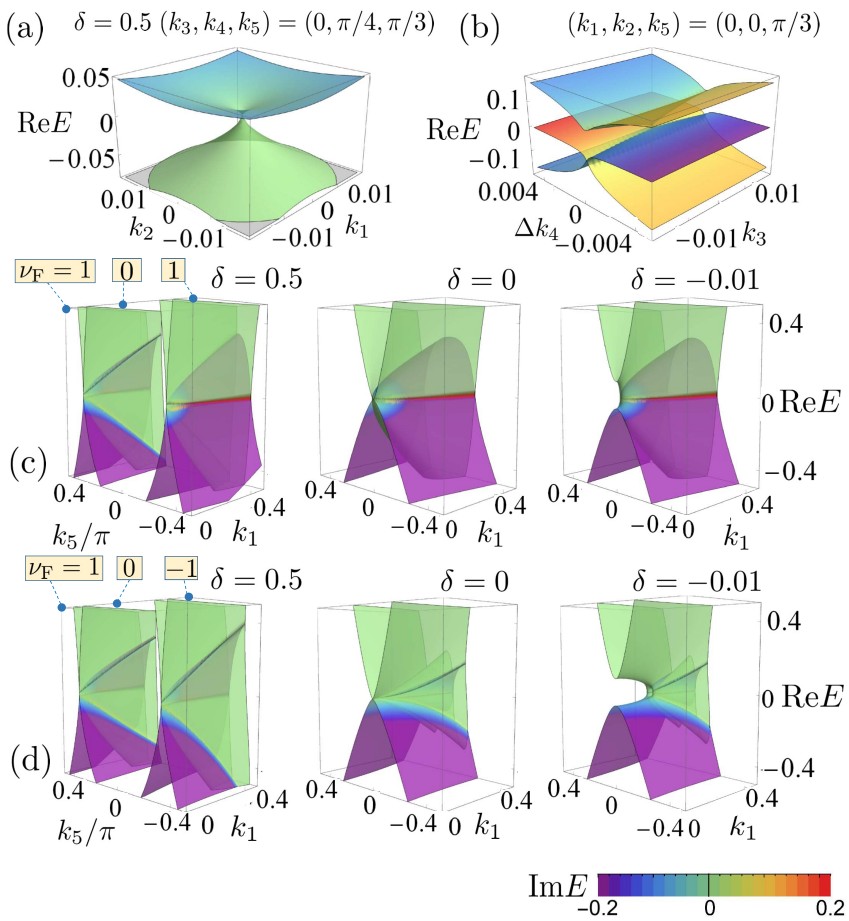

Figure 2: Energy bands of Hamiltonian (4) for $m_0 = 1.5$. The real (imaginary) part is represented as height (color). In panel (a), the complex conjugate of the upper band is omitted. Panels (a) and (b) are obtained for $k_5 = \pi/3$, $\delta = 0.5$, and $f(k_5) = 1$. Here, $\Delta k_4$ is defined as $\Delta k_4 = k_4 - \pi/2$. Panel (c) [(d)] displays pair annihilation of HEP3s for $(k_2, k_3, k_4) = (0, 0, \pi/2)$, and $f(k_5) = 1$ [$f(k_5) = 2\sin(k_5/2)$]. In these panels, numerically computed $\nu_F$ for $k_5 = -\pi/2$, 0, and $\pi/2$ at $\delta = 0.5$ is represented by numbers enclosed by blue squares. We used a mesh of $40^4$ points to evaluate the integrals with momenta $k_{1,2,3} \in [-\pi, \pi]$ and $k_4 \in [0, 2\pi]$.

We demonstrate the emergence of an HEP3 in five dimensions by analyzing a toy model whose Hamiltonian reads

$$H = \begin{pmatrix} 0 & 1 & 0 \\ 0 & 0 & 1 \\ \frac{\zeta_2}{6} & \zeta_1 & 0 \end{pmatrix}, \tag{4}$$

where the functions $\zeta_{1,2}(\boldsymbol{k})$ are parameterized by $\delta$, $m_0$, and $f(k_5)$ (for the explicit form, see Appendix B.1). Here the function $f(k_5)$ is either even [$f(k_5) = 1$] or odd [$f(k_5) = 2\sin(k_5/2)$].

To motivate the Hamiltonian form in Eq. (4), recall that a three-fold exceptional point is generally captured by a $3 \times 3$ Jordan block form [80]. The specified perturbation of the Jordan form in the bottom row of the Hamiltonian ensures that the resultants [recall Eq. (1)] are expressed as $r_j \propto \zeta_j$ for $j \in \{1, 2\}$ [77]. The sought higher Hopf topology is then ensured by imposing the appropriate dependence [107] of the functions $\zeta_{1,2}(\boldsymbol{k})$ on momenta (parameters) $\boldsymbol{k}$. The vanishing resultant vector determines where the HEP3s emerge in the momentum

space. The explicit form of the resultant vector [see Appendix B.1] indicates the emergence of HEP3s at $\boldsymbol{k} = (0, 0, 0, \pi/2, \pm\pi/3)$ for $\delta = 0.5$ and $m_0 = 1.5$. Figure 2(a,b) displays the emergence of the HEP3 at $\boldsymbol{k} = (0, 0, 0, \pi/2, \pi/3)$.

Notably, the HEP acts as its own antiparticle as a direct consequence of its $\mathbb{Z}_2$ topology. This fact is elucidated by examining pair annihilation of HEP3s in two cases: $f(k_5) = 1$ and $f(k_5) = 2\sin(k_5/2)$ [see Fig. 2(c,d)]. For $f(k_5) = 1$ and $\delta = 0.5$, the system hosts two HEP3s demarcated by the planes at $k_5 = \pi/2$, 0 and $-\pi/2$ where numerically computed $\nu_F$ is equal to 1, 0, and 1, respectively [for computation of $\nu_F$, see Appendix B.1]. As $\delta$ decreases, the two HEP3s approach and annihilate each other [see Fig. 2(c)]. Changing the parity of $f(k_5)$ flips the sign of the numerically computed $\nu_F$ at $k_5 = -\pi/2$. Even in this case, pair annihilation occurs [see Fig. 2(d)]. The occurrence of pair annihilation in both arrangements manifests that HEP3s are indeed protected by $\mathbb{Z}_2$ topology, implying that an HEP3 is its own antiparticle.

## 3 Symmetry-protected HEP5s with $\mathbb{Z}_2$ topology

Symmetry further enriches Hopf exceptional points, as exemplified by the emergence of symmetry-protected HEP5 in five dimensions. We consider a five-band non-Hermitian Hamiltonian which preserves parity-time ($PT$-) symmetry

$$U_{\mathrm{PT}} H^*(\boldsymbol{k}) U_{\mathrm{PT}}^{-1} = H(\boldsymbol{k}) \tag{5}$$

with a unitary matrix satisfying $U_{\mathrm{PT}} U_{\mathrm{PT}}^* = \mathbb{1}$. Here, $\mathbb{1}$ is the identity matrix, and asterisks denote complex conjugation. The $PT$-symmetry imposes the constraint

$$P(E) = P^*(E^*), \tag{6}$$

indicating that all coefficients of the characteristic polynomial $P(E)$ are real at each $\boldsymbol{k}$. For $n = 5$, the formation of an EP5 is captured by vanishing resultants $r_{1,\dots,4} = 0$ [see Eq. (1)]. Due to $PT$-symmetry [see Eqs. (5) and (6)] these resultants are real: $r_{1,\dots,4} \in \mathbb{R}$.

To expose whether the EP5 is in fact an HEP5, we consider a 4-sphere $S^4 \subset \mathbb{R}$ enclosing the band touching. Assuming $r_{1,\dots,4} \neq 0$ on the 4-sphere, we introduce a normalzied vector $\boldsymbol{n} = \boldsymbol{R}/\|\boldsymbol{R}\|$ with

$$\boldsymbol{R} = (r_1, r_2, r_3, r_4)^{\mathrm{T}}. \tag{7}$$

The normalized vector $\boldsymbol{n}$ defines a map from $S^4$ to $S^3$ whose topology is classified by $\pi_4(S^3) = \mathbb{Z}_2$. When the map of $\boldsymbol{n}$ possesses nontrivial $\mathbb{Z}_2$ topology, the enclosed band touching is a symmetry-protected HEP5. The topological invariant is introduced in a similar way as Eq. (3) with the only difference that we compute the $\mathbb{Z}_2$ invariant from the resultant vector in Eq. (7) instead of the one in Eq. (2).

We demonstrate the emergence of a symmetry-protected HEP5 by analyzing a toy model whose Hamiltonian reads

$$H = \begin{pmatrix} 0 & 1 & 0 & 0 & 0 \\ 0 & 0 & 1 & 0 & 0 \\ 0 & 0 & 0 & 1 & 0 \\ 0 & 0 & 0 & 0 & 1 \\ \frac{\zeta_4}{(5!)^3} & \frac{\zeta_3}{(5!)^2} & \frac{\zeta_2}{5!2!} & \frac{\zeta_1}{3!} & 0 \end{pmatrix} \tag{8}$$

with real functions $\zeta_{1,\dots,4}(\boldsymbol{k})$ parameterized by $\delta$, $m_0$, and $f(k_5)$ [for the explicit form, see Appendix B.2]. Here, $f(k_5)$ is either $f(k_5) = 1$ or $f(k_5) = 2\sin(k_5/2)$. This Hamiltonian preserves $PT$-symmetry [Eq. (5)] with $U_{\mathrm{PT}} = \mathbb{1}$. In analogy with the discussion of the Hamiltonian

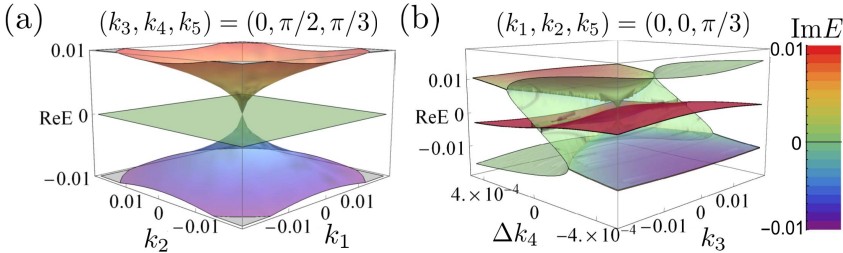

Figure 3: Energy bands of Hamiltonian (8) for $f(k_5) = 1$, $k_5 = \pi/3$, $m_0 = 1.5$ and $\delta = 0.5$. The real (imaginary) part is represented as height (color). Panel (a) [(b)] displays the data for $(k_3, k_4) = (0, \pi/2)$ [$(k_1, k_2) = (0, 0)$]. In panel (a), the complex conjugates of the upper and the lower bands are omitted. In panel (b), $\Delta k_4$ is defined as $\Delta k_4 = k_4 - \pi/2$.

in Eq. (4), the perturbation of the $5 \times 5$ Jordan form in Eq. (8) is chosen so that the resultants obey $r_j \propto \zeta_j$ for each $j \in \{1, \ldots, 4\}$. The explicit form of $\zeta_j(\boldsymbol{k})$ follows from a known representative of the nontrivial class in $\pi_4(S^3)$ [107].

The vanishing resultant vector $\boldsymbol{R} \propto (\zeta_1, \zeta_2, \zeta_3, \zeta_4)^{\mathrm{T}} = \boldsymbol{0}$ specifies where the symmetry-protected HEP5s emerge. The explicit form of the resultant vector [see Appendix B.2] indicates that the model in Eq. (8) hosts symmetry-protected $\mathbb{Z}_2$ HEP5s at $\boldsymbol{k} = (0, 0, 0, \pi/2, \pm\pi/3)$ which possesses $\mathbb{Z}_2$ topological charge [see Fig. 3].

The argument of $PT$-symmetry protected HEPs can be applied to other cases of symmetry: pseudo-Hermiticity, $CP$-, and chiral symmetry. We consider a Hamiltonian preserving pseudo-Hermiticity

$$U_{\mathrm{pH}} H(\boldsymbol{k}) U_{\mathrm{pH}}^\dagger = H^\dagger(\boldsymbol{k}) \tag{9}$$

with $U_{\mathrm{pH}}^\dagger$ being a unitary matrix and dagger denotes Hermitian conjugation. Because the transposition does not affect the determinant, Eq. (9) leads to Eq. (6), implying that the resultants [Eq. (1)] are real.

For $CP$- and chiral symmetry, the Hamiltonian obeys

$$U_{\mathrm{CP}} H^*(\boldsymbol{k}) U_{\mathrm{CP}}^\dagger = -H(\boldsymbol{k}), \tag{10}$$

$$U_\Gamma H^\dagger(\boldsymbol{k}) U_\Gamma^\dagger = -H(\boldsymbol{k}) \tag{11}$$

with $U_{\mathrm{CP}}$ and $U_\Gamma$ being unitary matrices. In these cases, replacing $H$ to $H' = iH$ reduces to the case of $PT$-symmetry or pseudo-Hermiticity. Therefore, symmetry-protected HEP$n$'s may emerge when systems preserve pseudo-Hermiticity, $CP$-, or chiral symmetry.

## 4 Symmetry-protected HEP4s with $\mathbb{Z}$ topology

Symmetry protection also enables HEP4s with $\mathbb{Z}$ topology. To illustrate such a possibility, we consider a four-band non-Hermitian Hamiltonian with $PT$-symmetry [see Eq. (5)] in a four-dimensional momentum space described by $\boldsymbol{k} = (k_1, k_2, k_3, k_4)$. For $n = 4$, the formation of an EP4 is captured by vanishing resultants $r_{1,\ldots,3} = 0$ [see Eq. (1)].

To expose whether the EP4 is in fact an HEP4, we consider a 3-sphere $S^3 \subset \mathbb{R}^4$ enclosing the band touching. The homotopy group $\pi_3(S^2) = \mathbb{Z}$ implies the existence of nontrivial maps $\boldsymbol{n}(\boldsymbol{k}) = \boldsymbol{R}/\|\boldsymbol{R}\|$. When the map of $\boldsymbol{n}$ possesses nontrivial $\mathbb{Z}$ topology, the enclosed band touching is a symmetry-protected HEP4.

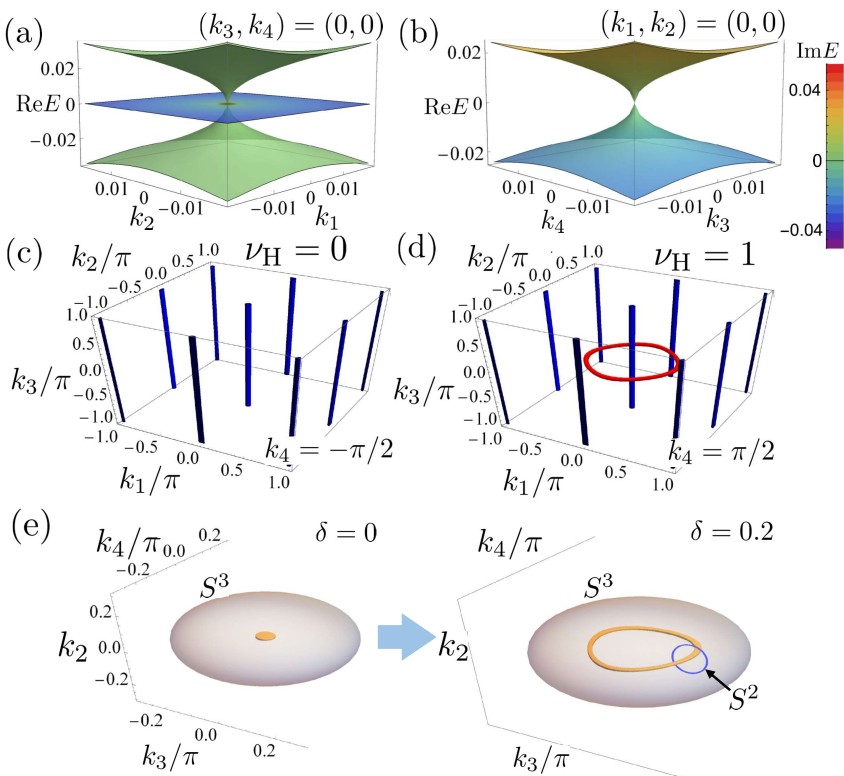

Figure 4: (a) and (b): Energy eigenvalues of Hamiltonian in Eq. (13) for $m_0 = 3$ and $\delta = 0$. The real (imaginary) part is represented as height (color). In panel (b), complex conjugates of the upper and lower bands are omitted. (c) [(d)]: Lines in the momentum space $(k_1, k_2, k_3)$ for $m_0 = 3$, $\delta = 0$ and $k_4 = -\pi/2$ [$k_4 = \pi/2$], where red and blue lines denote the momenta satisfying $\boldsymbol{R} \propto (0,0,1)^{\mathrm{T}}$ and $\boldsymbol{R} \propto (0,0,-1)^{\mathrm{T}}$, respectively. The linking of these lines determines the value of the Hopf invariant $\nu_{\mathrm{H}}$. (e): Momenta satisfying $\boldsymbol{R}(\boldsymbol{k}) = 0$ for $k_1 = 0$ and $\delta = 0$ resp. $\delta = 0.2$ (orange manifolds). As $\delta$ is introduced, the symmetry-protected HEP4 inflates into a loop. The gray oval and the blue loop illustrate the $S^3$ resp. $S^2$, both extending in the fourth dimension $k_1$ (not shown), on which one computes the Hopf invariant $\nu_{\mathrm{H}}$ resp. the resultant winding number $W_2$ (see Appendix A.2) [77].

The topology of such HEP4s is characterized by the Hopf invariant, expressed as [108–110]

$$\nu_{\mathrm{H}} = \oint \frac{d^3 \boldsymbol{p}}{2} \epsilon^{\mu\nu\rho} A_\mu F_{\nu\rho}, \tag{12}$$

with $\mu, \nu, \rho = 1, 2, 3$. Here, $A_\mu$ and $F_{\mu\nu}$ are obtained from the resultant Hamiltonian (for the definitions, see Appendix A.3). Vector $\boldsymbol{p}$ parametrizes the 3-sphere in the momentum space.

We demonstrate the emergence of a symmetry-protected HEP4 by analyzing a toy model whose Hamiltonian reads

$$H = \begin{pmatrix} 0 & 1 & 0 & 0 \\ 0 & 0 & 1 & 0 \\ 0 & 0 & 0 & 1 \\ \frac{\zeta_3}{24^2} & -\frac{\zeta_2}{24} & \frac{\zeta_1}{2} & 0 \end{pmatrix}, \tag{13}$$

with real functions $\zeta_{1,\dots,3}(\boldsymbol{k})$ parametrized by $\delta$ and $m_0$ [for the explicit form, see Appendix B.3].

As is the case with Hamiltonians in Eqs. (4) and (8), the Hopf topology [108] of $\zeta$'s in the perturbed model [Eq. (13)] is equivalent to that of the resultant vector.

The vanishing resultant vector $\boldsymbol{R} \propto (\zeta_1, \zeta_2, \zeta_3)^{\mathrm{T}} = \boldsymbol{0}$ specifies where the symmetry-protected HEP4 emerges in the momentum space. The explicit form of the resultant [see Appendix B.3] indicates the emergence of a symmetry-protected HEP4 at $\boldsymbol{k} = \boldsymbol{0}$ for $m_0 = 3$ [see Fig. 4(a,b)]. Here, we characterize the symmetry-protected HEP4 enclosed by two planes at $k_4 = -\pi/2$ and $k_4 = \pi/2$. Numerically evaluating the Hopf invariant [Eq. (12)], we obtain $\nu_{\mathrm{H}} = 0$ [$\nu_{\mathrm{H}} = 1$] for $k_4 = -\pi/2$ [$k_4 = \pi/2$] which is consistent with linking of the inverse maps in the momentum space [see Fig. 4(c,d)] [111, 112]. These results indicate that the symmetry-protected HEP4 is characterized by $\nu_{\mathrm{H}} = 1$.

## 5 General characteristics of HEPs

### 5.1 Multiply-charged aspect of HEPs

Reference 78 has pointed out that the codimension of symmetry-protected EP4s is three, whereas the codimension of the HEP4 in our model is four [see Fig. 4(a,b)]. This mismatch in codimension implies that the HEP4 inflates into a loop of EP4s under a generic perturbation of the Hamiltonian matrix [see Fig. 4(e)]. Such a perturbation does not trivialize the Hopf topology but enriches it [29]: the loop of EP4s carries both the Hopf invariant on $S^3$ and the resultant winding number on $S^2$. In the same spirit, a similar assignment of multiple topological invariants applies to all HEPs introduced in this work.

### 5.2 Additional bands

So far, we have discussed HEP$n$'s in $n$-band models. For systems with more than $n$ bands, EP$n$'s are still associated with winding of the resultant vector; however, the converse does not hold. Specifically, as discussed in Appendix C, it is possible to find situations where a finite value of the winding number leads to the vanishing resultant vector $\boldsymbol{R} = \boldsymbol{0}$ without a symmetry-protected EP3.

Nevertheless, topological invariants of resultants are applicable when we focus around an HEP$n$ (or an EP$n$). Specifically, if we know that an HEP$n$ emerges at momenta $\boldsymbol{k} = \boldsymbol{k}_0$ and energy $E = E_0$, we may apply the Taylor expansion to the characteristic polynomial $P(E) = \det[H(\boldsymbol{k}) - E\mathbb{1}]$, which leads to a polynomial $\tilde{P}(\tilde{E})$ with degree $n$

$$\tilde{P}(\tilde{E}) = \sum_{j=0}^{n} a_j(\boldsymbol{k})\tilde{E}^j, \tag{14}$$

with $a_j(\boldsymbol{k})$ ($j = 0, 1, \ldots, n$) being complex functions and $\tilde{E} = E - E_0$. Computing the topological invariant [e.g., Eq. (3)] of $\tilde{P}(\tilde{E})$ around $\boldsymbol{k} = \boldsymbol{k}_0$, we can characterize the HEP$n$ for systems with more than $n$ bands.

### 5.3 Higher dimensions

Higher homotopy groups of spheres indicate abundant topology of HEP$n$'s. HEP$n$'s are captured by the $(2n-2)$-component resultant vector, and thus are classified by $\pi_{c-1}(S^{2n-3})$ with codimension $c \geq 4$. Symmetry-protected HEP$n$'s are captured by the $(n-1)$-component resultant vector, and thus are classified by $\pi_{c-1}(S^{n-2})$. The classification results for HEP$n$'s ($n = 3, 4$) and symmetry-protected HEP$n$'s ($n = 4, 5, 6, 7$) are summarized in Table 1 (see also Sec. 4.1 of Ref. [113] and Chapter XIV of Ref. [114]). This table predicts the presence of various HEP$n$'s

| $c$ | HEP3 SP-HEP4, SP-HEP5 | SP-HEP6 | HEP4 SP-HEP7 |
|---|---|---|---|
| 4 | $\mathbb{Z}$ | 0 | 0 |
| 5 | $\mathbb{Z}_2$ | $\mathbb{Z}$ | 0 |
| 6 | $\mathbb{Z}_2$ | $\mathbb{Z}_2$ | $\mathbb{Z}$ |
| 7 | $\mathbb{Z}_{12}$ | $\mathbb{Z}_2$ | $\mathbb{Z}_2$ |
| 8 | $\mathbb{Z}_2$ | $\mathbb{Z} \times \mathbb{Z}_{12}$ | $\mathbb{Z}_2$ |
| 9 | $\mathbb{Z}_2$ | $\mathbb{Z}_2^2$ | $\mathbb{Z}_{24}$ |
| 10 | $\mathbb{Z}_3$ | $\mathbb{Z}_2^2$ | $\mathbb{Z}_2$ |
| 11 | $\mathbb{Z}_{15}$ | $\mathbb{Z}_{24} \times \mathbb{Z}_3$ | $\mathbb{Z}_2$ |
| 12 | $\mathbb{Z}_2$ | $\mathbb{Z}_{15}$ | $\mathbb{Z}_2$ |
| 13 | $\mathbb{Z}_2^2$ | $\mathbb{Z}_2$ | $\mathbb{Z}_{30}$ |
| 14 | $\mathbb{Z}_{12} \times \mathbb{Z}_2$ | $\mathbb{Z}_2^3$ | $\mathbb{Z}_2$ |
| 15 | $\mathbb{Z}_{84} \times \mathbb{Z}_2^2$ | $\mathbb{Z}_{120} \times \mathbb{Z}_{12} \times \mathbb{Z}_2$ | $\mathbb{Z}_2^3$ |
| 16 | $\mathbb{Z}_2^2$ | $\mathbb{Z}_{84} \times \mathbb{Z}_2^5$ | $\mathbb{Z}_{72} \times \mathbb{Z}_2$ |

Table 1: Hopf exceptional points (HEPs) are classified by higher homotopy groups of spheres. The topology explicitly analyzed in this manuscript corresponds to the blue entries. HEP$n$'s [symmetry-protected HEP$n$'s] of codimension $c = 4, 5, \ldots$ are classified by $\pi_{c-1}(S^{2n-3})$ $[\pi_{c-1}(S^{n-2})]$.

following exotic fusion rules. For instance, $\mathbb{Z}_3$ topology of codimension $c = 10$ implies an HEP3 annihilate with two copies of itself, which is reminiscent of parafermions [115–118]. In addition, there exist HEP3s with $\mathbb{Z}_{12}$ topology of codimension $c = 7$ and HEP4s with $\mathbb{Z}_{24}$ topology of codimension $c = 9$.

# 6 Conclusion and outlook

We have discovered novel non-Hermitian topological excitations, dubbed Hopf exceptional points. Saliently, HEP3s and symmetry-protected HEP5s exhibit an unusual $\mathbb{Z}_2$ topology, meaning that they act as their own antiparticle. This striking feature arises from the homotopy group $\pi_4(S^3) = \mathbb{Z}_2$. We have further discovered symmetry-protected HEP4s whose topology is classified by $\pi_3(S^2) = \mathbb{Z}$. Leveraging higher homotopy groups of spheres, we elucidate the potential presence of HEP$n$'s characterized by abundant finite groups (e.g., $\mathbb{Z}_3$, $\mathbb{Z}_{12}$, and $\mathbb{Z}_{24}$) beyond the classification table of Bernard-LeClair symmetry classes.

Our work on non-Hermitian multiband systems opens up a new direction of topological physics. Here, we outline several concrete open questions motivated by our findings. On one hand, from the implementation perspective, we anticipate that HEP$n$'s can be realized in a variety of experimental settings. Notably they may be realized in terms of non-unitary photon dynamics where both the eigenvalues and eigenstates have been measured accurately at and around multifold EPs [93], hence suggesting a path to a particularly comprehensive simulation of HEP$n$'s. Further promising platforms include nitrogen-vacancy spin systems [103] and coupled micro-resonators [94] in which multifold EPs have also been realized. Quite generally, the high controllability of metamaterials allows access to momentum (or parameter) spaces with dimensions larger than three [61, 62, 75–77], thus inviting various realistic experimental verifications of the unusual topology of HEP$n$'s.

On the other hand, our work also indicates several concrete theoretical aspects that deserve a deeper mathematical analysis. First, Table 1 implies the presence of novel HEPs following

a unique fusion rule due to their topology. Explicit analysis based on topological invariants and concrete models is an interesting issue. Next, unless an explicit band structure is provided, one-to-one correspondence between EP$n$'s and the resultant winding numbers is lost for many-band systems. This observation calls for a more general topological characterization Hopf exceptional points. Finally, it is interesting to consider non-Hermitian topological bands, captured by suitably adapted Hopf invariants [112], that arise in models with a hopfion texture [119] of the resultant vector over the Brillouin zone torus. We postpone the analysis of these questions to future studies.

## Acknowledgments

T. Y. is grateful for the support from the ETH Pauli Center for Theoretical Studies.

**Funding information** T. Y. is supported by JSPS KAKENHI Grant Nos. JP21K13850, JP23KK0247, JP25K07152, and JP25H02136 as well as JSPS Bilateral Program No. JPJSBP120249925 and Yamada Science Foundation. E. J. B. is supported by the Wallenberg Scholars program (2023.0256) and the Göran Gustafsson Foundation for Research in Natural Sciences and Medicine. T. B. was supported by the Starting Grant No. 211310 by the Swiss National Science Foundation (SNSF).

## A  Defining topology from resultants

### A.1  Resultant of a pair of polynomials

For given two polynomials

$$f(x) = a_n x^n + \ldots + a_1 x + a_0, \tag{A.1}$$
$$g(x) = b_m x^m + \ldots + b_1 x + b_0, \tag{A.2}$$

with complex coefficients $a$'s and $b$'s, the resultant is defined as

$$\mathrm{Res}[f(x), g(x)] = a_n^m b_m^n \prod_{i,j} (\alpha_i - \beta_j). \tag{A.3}$$

Here, $\alpha$'s and $\beta$'s are roots of polynomials $f(x)$ and $g(x)$, respectively. Symbol $\prod_{i,j}$ denotes the product of all pairs of the roots. The resultant vanishes when the two polynomials have a common root.

The resultant can be computed from the Sylvester matrix

$$\mathrm{Res}[f(x), g(x)] = \det \begin{bmatrix} a_n & \cdots & a_0 & & & \\ & a_n & \cdots & a_0 & & \\ & & \ddots & & \ddots & \\ & & & a_n & \cdots & a_0 \\ b_m & \cdots & b_0 & & & \\ & b_m & \cdots & b_0 & & \\ & & \ddots & & \ddots & \\ & & & b_m & \cdots & b_0 \end{bmatrix}. \tag{A.4}$$

Here, the empty elements are zero. The size of the matrix is $n + m$ and the first $n$ rows are composed of the coefficients $a$'s and the remaining $m$ rows are composed of the coefficients $b$'s.

## A.2 Resultant winding number

We next consider an $N$-component resultant vector $\boldsymbol{R}(\boldsymbol{k})$, such as the one in Eq. (2), defined on an $N$-dimesional momentum space $\mathbb{R}^N$. When the norm is finite $\|\boldsymbol{R}\| \neq 0$ on an $(N-1)$-sphere $S^{N-1} \subset \mathbb{R}^N$, one can consider the normalized vector $\boldsymbol{n}(\boldsymbol{k}) = \boldsymbol{R}(\boldsymbol{k})/\|\boldsymbol{R}(\boldsymbol{k})\|$ whose topology is classified by $\pi_{N-1}(S^{N-1}) = \mathbb{Z}$.

The topology of such a map is characterized by the resultant winding number $W_{N-1}$ [77,78]

$$W_{N-1} = \frac{\epsilon^{i_1 \cdots i_N}}{A_{N-1}} \int d^{N-1} \boldsymbol{p} f_{i_1 \cdots i_N}(\boldsymbol{p}), \tag{A.5}$$

$$f_{i_1 \cdots i_N}(\boldsymbol{p}) = n_{i_1} \partial_1 n_{i_2} \partial_2 n_{i_3} \cdots \partial_{N-1} n_{i_N}, \tag{A.6}$$

where the integral is taken over $S^{N-1}$ in the momentum space parameterized by vector $\boldsymbol{p}$. Here, the area of the $(N-1)$-dimensional sphere $A_{N-1}$ is expressed by

$$A_{2l-1} = \frac{2\pi^l}{(l-1)!}, \tag{A.7}$$

$$A_{2l-2} = \frac{2^{2l-1}\pi^{l-1}(l-1)!}{(2l-2)!}, \tag{A.8}$$

with a positive integer $l$.

The above invariant is rewritten as Chern numbers or winding numbers of the resultant Hamiltonian [78]. Specifically, for $N = 3$, Eq. (A.5) is rewritten as the first Chern number of the resultant Hamiltonian $H_R(\boldsymbol{k}) = \boldsymbol{R} \cdot \boldsymbol{\sigma}$. For $N = 4$, Eq. (A.5) is rewritten as the winding number of three-dimensional chiral symmetric Hamiltonian $H_R = \sum_{i=1,\dots,4} R_i \gamma_i$ satisfying $\gamma_5 H_R \gamma_5 = -H_R$ with $\gamma = (\sigma_1 \tau_0, \sigma_2 \tau_0, \sigma_3 \tau_1, \sigma_3 \tau_2, \sigma_3 \tau_3)^{\mathrm{T}}$. Here, $\tau_{1,\dots,3}$ are Pauli matrices, and $\tau_0$ is the $2 \times 2$ identity matrix. We note that the presence of the gap of $H_R$ on the sphere is reduced to $\|\boldsymbol{R}\| \neq 0$. This is because the eigenvalues are given by $E_R = \pm\|\boldsymbol{R}\|$ which arises from the anti-commutation relation $\{\gamma_i, \gamma_j\} = 2\delta_{ij}$.

## A.3 Berry connection and curvature of resultant Hamiltonians for Hopf topology

The $\mathbb{Z}_2$ invariant in Eq. (3) is obtained from the resultant vector $\boldsymbol{R} = (R_1, R_2, R_3, R_4)^{\mathrm{T}}$ $(R_{1,\dots,4} \in \mathbb{R})$. Specifically, the Berry connection $A_\mu$ and the Berry curvature $F_{\mu\nu}$ are defined as

$$A_\mu = \frac{1}{2\pi\mathrm{i}} \langle z | \partial_\nu z \rangle, \tag{A.9}$$

$$F_{\mu\nu} = \frac{1}{2\mathrm{i}} \big( \langle \partial_\mu z | \partial_\nu z \rangle - \langle \partial_\nu z | \partial_\mu z \rangle \big), \tag{A.10}$$

with the negative eigenstate $|z\rangle$ of the resultant Hamiltonian $\tilde{\boldsymbol{n}} \cdot \boldsymbol{\sigma}$ with $\tilde{\boldsymbol{n}} = \tilde{\boldsymbol{R}}/\|\tilde{\boldsymbol{R}}\|$ and $\tilde{\boldsymbol{R}} = (R_1, R_2, R_3)$. Here, $\sigma_{1,\dots,3}$ denote Pauli matrices. The ratio $\|\tilde{\boldsymbol{R}}\|/R_4$ defines the phase $\Delta\varphi(\boldsymbol{p}) = 2\varphi(\boldsymbol{p})$ with $\varphi = \arctan\|\tilde{\boldsymbol{R}}\|/R_4$ ($0 \leq \varphi \leq \pi$). Substituting $\Delta\varphi$, $A_\mu$, and $F_{\mu\nu}$ into Eq. (3) yields the $\mathbb{Z}_2$ invariant $\nu_F$.

The $\mathbb{Z}$ invariant in Eq. (12) is also obtained from the resultant vector $\boldsymbol{R} = (R_1, R_2, R_3)^{\mathrm{T}}$ $(R_{1,\dots,3} \in \mathbb{R})$. Specifically, the Berry connection $A_\mu$ and the Berry curvature $F_{\mu\nu}$ are defined in the same way as Eqs. (A.9) and (A.10) except for the definition of $|z\rangle$. In this case, $|z\rangle$ is defined as the negative eigenstate of the resultant Hamiltonian $\boldsymbol{n} \cdot \boldsymbol{\sigma}$ with $\boldsymbol{n} = \boldsymbol{R}/\|\boldsymbol{R}\|$. Substituting the specific form of the Berry curvature $A_\mu$ and the Berry connection $F_{\mu\nu}$ into Eq. (12) yields $\mathbb{Z}$ invariant $\nu_H$.

# B    Details of the presented Hamiltonians

## B.1    Model in Eq. (4) and computation of the $\mathbb{Z}_2$ invariant

The explicit form of $\zeta$'s is

$$\zeta_1 = -2\sin\phi\left(\eta_\uparrow^* \eta_\downarrow\right), \tag{B.1a}$$

$$\zeta_2 = -\sin\phi\left(|\eta_\uparrow|^2 - |\eta_\downarrow|^2\right) + i\cos\phi. \tag{B.1b}$$

Here $\eta_\uparrow$, $\eta_\downarrow$ and $\phi$ are defined as

$$\eta_\uparrow = \sin k_1 + i f(k_5)\sin k_2, \tag{B.2a}$$

$$\eta_\downarrow = \sin k_3 + i\left[\xi(\boldsymbol{k}) + \frac{3}{2}\sin k_4 - 3(\cos k_5 + \delta)\right], \tag{B.2b}$$

$$\phi = \frac{\pi}{2}(1 - \cos k_4), \tag{B.2c}$$

with $\xi(\boldsymbol{k}) = \sum_{j=1,2,3}\cos k_j - m_0$ and $f(k_5)$ being $f(k_5) = 1$ or $f(k_5) = 2\sin(k_5/2)$.

The resultant vector $\boldsymbol{R}$ is obtained as

$$\boldsymbol{R} = (3!)^2\left(-2\sin\phi\,\mathrm{Re}[\eta_\uparrow\eta_\downarrow], -2\sin\phi\,\mathrm{Im}[\eta_\uparrow\eta_\downarrow], -\sin\phi(|\eta_\uparrow|^2 - |\eta_\downarrow|^2), \cos\phi\right)^{\mathrm{T}}. \tag{B.3}$$

The above equation indicates that the resultant vector vanishes when $\eta_\uparrow = \eta_\downarrow = 0$ and $\phi = \pi/2$ both hold.

From the given resultant vector [see Eq. (B.3)], the $\mathbb{Z}_2$ invariant $\nu_\mathrm{F}$ is obtained as follows. With $\tilde{\boldsymbol{n}} = \tilde{\boldsymbol{R}}/\|\tilde{\boldsymbol{R}}\|$ and Eq. (B.3), the resultant Hamiltonian is obtained as

$$\tilde{\boldsymbol{n}}\cdot\boldsymbol{\sigma} = -\frac{1}{\sqrt{|\eta_\uparrow|^2 + |\eta_\downarrow|^2}}\begin{pmatrix} |\eta_\uparrow|^2 - |\eta_\downarrow|^2 & 2\eta_\uparrow\eta_\downarrow^* \\ 2\eta_\uparrow^*\eta_\downarrow & -|\eta_\uparrow|^2 + |\eta_\downarrow|^2 \end{pmatrix}. \tag{B.4}$$

Thus, the negative eigenstate of $\tilde{\boldsymbol{n}}\cdot\boldsymbol{\sigma}$ is obtained as

$$|z(\boldsymbol{k})\rangle = \frac{1}{\sqrt{|\eta_\uparrow|^2 + |\eta_\downarrow|^2}}\begin{pmatrix} \eta_\uparrow \\ \eta_\downarrow \end{pmatrix}. \tag{B.5}$$

In addition, from the resultant vector in Eq. (B.3), we obtain

$$\Delta\varphi = 2\arctan\|\tilde{\boldsymbol{R}}\|/R_4 = 2\phi. \tag{B.6}$$

Substituting the obtained $|z\rangle$ and $\Delta\varphi$ into Eq. (3), we can numerically compute the $\mathbb{Z}_2$ invariant.

## B.2    Model in Eq. (8)

The explicit form of $\zeta$'s are

$$\zeta_1 + i\zeta_2 = -2\sin\phi(\eta_\uparrow\eta_\downarrow), \tag{B.7a}$$

$$\zeta_3 = -\sin\phi(|\eta_\uparrow|^2 - |\eta_\downarrow|^2), \tag{B.7b}$$

$$\zeta_4 = \cos\phi, \tag{B.7c}$$

with $\eta$'s and $\phi$ defined in Eq. (B.2).

The resultant vector of this Hamiltonian is obtained as

$$\boldsymbol{R} = (5!)^2\left(-2\sin\phi\,\mathrm{Re}[\eta_\uparrow\eta_\downarrow], -2\sin\phi\,\mathrm{Im}[\eta_\uparrow\eta_\downarrow], -\sin\phi(|\eta_\uparrow|^2 - |\eta_\downarrow|^2), \cos\phi\right)^{\mathrm{T}}, \tag{B.8}$$

which is proportional to the resultant vector in Eq. (B.3). The above equation indicates that the resultant vector vanishes when $\eta_\uparrow = \eta_\downarrow = 0$ and $\phi = \pi/2$ both hold.

## B.3 Model in Eq. (13)

The explicit form of $\zeta$'s are

$$\zeta_1 + i\zeta_2 = 2(\eta_\uparrow^* \eta_\downarrow), \tag{B.9a}$$

$$\zeta_3 = |\eta_\uparrow|^2 - |\eta_\downarrow|^2 + \delta. \tag{B.9b}$$

Here $\eta_\uparrow$ and $\eta_\downarrow$ are

$$\eta_\uparrow = \sin k_1 + i \sin k_2, \tag{B.10a}$$

$$\eta_\downarrow = \sin k_3 + i \left[ \xi(\boldsymbol{k}) + \sin k_4 \right], \tag{B.10b}$$

with $\xi(\boldsymbol{k}) = \sum_{j=1,2,3} \cos k_j - m_0$.

The resultant vector $\boldsymbol{R}$ is obtained as

$$\boldsymbol{R} = -(24)^2 \left( 2\mathrm{Re}[\eta_\uparrow^* \eta_\downarrow], 2\mathrm{Im}[\eta_\uparrow^* \eta_\downarrow], |\eta_\uparrow|^2 - |\eta_\downarrow|^2 + \delta \right)^{\mathrm{T}}. \tag{B.11}$$

The above equation indicates that the resultant vector vanishes when $\eta_\uparrow = 0$ and $|\eta_\downarrow| = \sqrt{\delta}$ both hold for $\delta > 0$ or when $\eta_\downarrow = 0$ and $|\eta_\uparrow| = \sqrt{|\delta|}$ both hold for $\delta < 0$.

## C    Fake EP3s

For three-band systems, EP3s with $PT$-symmetry are characterized by the resultant winding number [77, 78] of the resultant Hamiltonian $R_1\sigma_1 + R_2\sigma_2$ with $\boldsymbol{R} = (r_1, r_2)$ which satisfies chiral symmetry. However, for systems with four or more bands, the resultant winding number may additionally capture "fake EP3s" unless Taylor expansion is applied. This is because the resultant vector may vanish without a triple root in the characteristic polynomial $P(E)$ whose degree is four or higher.

As an example, we consider a non-Hermitian Hamiltonian

$$H = \begin{pmatrix} 1 + ik_1 & 0 & 0 & 0 \\ 0 & 1 - ik_1 & 0 & 0 \\ 0 & 0 & ik_2 & 1 \\ 0 & 0 & 1 & -ik_2 \end{pmatrix}, \tag{C.1}$$

satisfying $PT$-symmetry [Eq. (5)] with

$$U_{\mathrm{PT}} = \begin{pmatrix} 0 & 1 & 0 & 0 \\ 1 & 0 & 0 & 0 \\ 0 & 0 & 1 & 0 \\ 0 & 0 & 0 & 1 \end{pmatrix}. \tag{C.2}$$

The resultant winding number is defined as

$$W_1 = \oint \frac{dp}{2\pi i} \partial_p \log[R_1(p) + iR_2(p)], \tag{C.3}$$

where $p$ parametrizes the circle in the momentum space. The winding number $W_1$ (see Appendix A.2) is finite for the path illustrated by the black arrow in Fig. 5(a). However, the system does not host EP3s [see Fig. 5(b)].

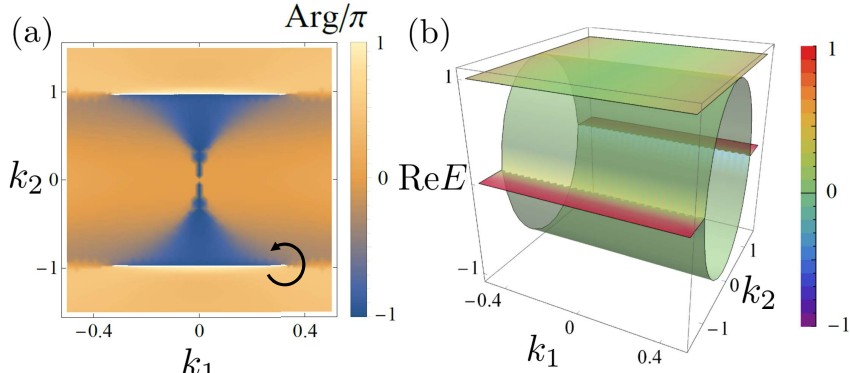

Figure 5: (a) [(b)]: Argument of $R_1 + iR_2$ [band structure] of Hamiltonian (C.1). In panel (b), the complex conjugate of the top band is omitted.

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
