# Peer review of "Hopf Exceptional Points"

_SciPost Physics_

## Round 1 · Referee Report · Anonymous (Referee 1) · 2025-8-4

Strengths

  • Describes an interesting type of exceptional point.
  • Provides specific models to illuminate key features.

Weaknesses

  • Lacks careful connection to existing literature.
  • Description of key features could be improved.

Report

This paper describes topological properties of a certain class of exceptional points embedded in a suitable parameter space. It is a timely contribution to a topical field, with the main drawback being the lack of connections to an ample body of existing literature. This also impacts on the assessment of its novelty, which should be reserved until the clarifications listed below are made.

Requested changes

1- The introduction and later parts of the paper mixes different topological notions - quasiparticles from fermionic Fock space topology and singularities of real and complex fields, such as diabolic points and exceptional points. See also phrases such as "meaning that they act as their own antiparticle...", "Notably, the HEP acts as its own antiparticle...". A quasiparticle is a very well-defined concept; it has a mass, can move, etc, and all of this does not apply here. On the other hand, static singularities of complex fields [i the mathematical sense] have well defined properties, too, and these apply very well to EPs, also in the variant described here. I understand the motivation being the annihilation property of the HEPs. However, a clear distinction of these only superficially overlapping concepts would add clarity. This also impacts on the description and understanding of the nature of HEPs and choice of models; see further below.

2- "However, topology of the formerly reported n-fold exceptional points (EPn’s) is generally characterized by Z invariants [77, 78]." This should make connection to existing EP topology as developed, eg, by the JW Ryu group, Phys. Rev. A 85, 042101 (2012); Phys. Rev. A 106, 012218 (2022); Commun. Phys. 7, 109 (2024) [possibly also other groups]. For instance, the last paper states in the abstract "our proposed classification gives rise to finer Z2 classifications depending on the presence of a π Berry phase after the encircling of the EPs." Please clarify, in detail, how this relates to the HEPs described here. The authors should not shy away to clarify if their results happen to perfectly fit into other such existing frameworks-the paper may still be very valuable in that case.

3- The notion of HEP would benefit from some important basic statements, especially around the introductory example (4). - It appears that in the parameter space of eta1 and eta2, the degeneracy is a normal EP3s; if so this should be clearly stated. - Are these EPs also HEPs in the enlarged parameter space including the control parameters used to make the two HEPs meet and annihilate (such as phi and delta in he model of Eq (4))? - In the context it should be clarified how much of this is specific to EPs and how much is true for singularities of a suitable m-dimensional complex field. It seems much of this is tied to the mapping of the parameters to eta1 and eta2, and not particular to that we are dealing with an EP. Would any other physical property requiring eta1=eta2=0 do? The statement "The sought higher Hopf topology is then ensured by imposing the appropriate dependence [107] of the functions ζ1,2(k) on momenta (parameters) k." seems to imply this, but I would prefer a more transparent statement, ideally already in the introduction. - For an analogously general perspective for a single (m=1) complex field see eg arXiv:2507.14373. - In this context, the choice of the models should be clarified. Is their form dictated by the HEP use case, or do they have other features making them preferable over alternative models that may already exist in the literature? - Can a simple picture for the HEP annihilation been given? It may be something as simple as this: Consider an EP2 with Arnold-Jordan normal form (0,1;eps,0) and eps=k^2+phi, real k and phi. Then EPs annihilate at phi=0, k=0; there are two EPs at phi<0 and none at phi>0. While this is (hopefully!) not an HEP, a description of such form provides a much clearer picture, e.g. about the role of the underlying parameter space. Does something similar apply to HEPs?

-4 "The specified perturbation of the Jordan form in the bottom row". The JNF is 0 in the bottom row; the authors use the Arnold Jordan normal form.

-5 The Arnold Jordan normal form, and possibly also the resultant approach, fail at EPs with geometric multiplicity g>1. Do HEPs with g>1 exist? For direct approach sidestepping the Arnold Jordan normal form and operating directly in parameter space of Hamiltonians see Physical Review Research 7 (2), 023062 (2025).

-6 The reference list [76–84] introducing higher-order EPs as a concept should focus on original references such as W D Heiss 2008 J. Phys. A: Math. Theor. 41 244010, plus what would come up in a forward-backward literature search (including [83] is fine), instead of the self citations. The references also leave out the important insight into EPn arrived from sensing, see Photon. Res. 8, 1457 (2020) and reference forward and backwards from there.

-7 "Reference 78 has pointed out that the codimension of symmetry-protected EP4s is three,..." It is known for a much longer time that EPns have codimension 2n-2 (it follows directly from the Arnold Jordan Normal form).

-8 Generally a more thorough approach to referencing in this field is required. This is not so much as expanding the list in this active field, but giving proper credit to the original works establishing key knowledge.

Recommendation

Ask for major revision

  • validity: high
  • significance: high
  • originality: good
  • clarity: ok
  • formatting: excellent
  • grammar: excellent

Author:  Tsuneya Yoshida  on 2025-10-29  [id 5966]

(in reply to Report 1 on 2025-08-04)
Category:
answer to question

Warnings issued while processing user-supplied markup:

  • Inconsistency: Markdown and reStructuredText syntaxes are mixed. Markdown will be used.
    Add "#coerce:reST" or "#coerce:plain" as the first line of your text to force reStructuredText or no markup.
    You may also contact the helpdesk if the formatting is incorrect and you are unable to edit your text.

Reply to Referee 1's comments

This paper describes topological properties of a certain class of exceptional points embedded in a suitable parameter space. It is a timely contribution to a topical field, with the main drawback being the lack of connections to an ample body of existing literature. This also impacts on the assessment of its novelty, which should be reserved until the clarifications listed below are made.

We would like to thank Referee 1 for their careful reading of our manuscript. We also appreciate their comments which were helpful in improving our manuscript. After carefully studying all the comments by the referee, we found that our insufficient explanations in the previous version prevented them from assessing the novelty. The novelty of our results is discovering a novel type of $n$-fold EPs (EP$n$) with $n\geq 3$ dubbed Hopf EP$n$s. In particular, Hopf EP3s and symmetry-protected HEP5s possess $\mathbb{Z}_2$ topology in contrast to formally reported EP$n$s. This point is supported by Referee B, stating "This article introduces a new class of exceptional points (EP), dubbed Hopf exceptional points. While previous topological classifications of exceptional points have, to my knowledge, only involved integer invariants Z, the exceptional points identified in this manuscript are instead characterized by finite groups such as Z2 and Z3, involving a very different robustness to continuous perturbations that may or may not annihilate them." and "I find this work highly pertinent and timely. It represents a very interesting extension of the existing topological classification of exceptional points. I therefore recommend it for publication." To make the novelty clear, we have substantially revised the manuscript by taking all the comments into account. We also attach our revised manuscript with the changes highlighted. In the following, we reply to them one by one.

Reply to comment: 1-[1]

1- The introduction and later parts of the paper mixes different topological notions - quasiparticles from fermionic Fock space topology and singularities of real and complex fields, such as diabolic points and exceptional points. See also phrases such as "meaning that they act as their own antiparticle...", "Notably, the HEP acts as its own antiparticle...". A quasiparticle is a very well-defined concept; it has a mass, can move, etc, and all of this does not apply here. On the other hand, static singularities of complex fields [i the mathematical sense] have well defined properties, too, and these apply very well to EPs, also in the variant described here. I understand the motivation being the annihilation property of the HEPs. However, a clear distinction of these only superficially overlapping concepts would add clarity. This also impacts on the description and understanding of the nature of HEPs and choice of models; see further below.

We thank the referee for pointing out the potential confusion between quasiparticles in the traditional condensed matter sense (quasiparticles with mass and mobility, etc.) and band touching (singularity of the fields). In our manuscript, the term "quasiparticle" is used in the broader sense common in topological condensed matter physics where it denotes topologically protected excitations or band singularities, irrespective of whether they have a conventional particle-like dispersion. This broader definition is widely used in the literature, e.g., for Weyl points and Majorana zero modes [see references Rev. Mod. Phys. 82, 3045 (2010); Rev. Mod. Phys. 83, 1057 (2011); Science 353, 6299 (2016)]. Such topologically protected "quasiparticles" include HEP$n$s as well as exceptional points. To avoid ambiguity, we add a footnote in the first paragraph of Sec. 1 explaining that "quasiparticle" in our manuscript refers to a topological quasiparticle which means a topologically protected excitation or band touching.

Reply to comment: 1-[2]

2- "However, topology of the formerly reported n-fold exceptional points (EPn’s) is generally characterized by Z invariants [77, 78]." This should make connection to existing EP topology as developed, eg, by the JW Ryu group, Phys. Rev. A 85, 042101 (2012); Phys. Rev. A 106, 012218 (2022); Commun. Phys. 7, 109 (2024) [possibly also other groups]. For instance, the last paper states in the abstract "our proposed classification gives rise to finer Z2 classifications depending on the presence of a $\pi$ Berry phase after the encircling of the EPs." Please clarify, in detail, how this relates to the HEPs described here. The authors should not shy away to clarify if their results happen to perfectly fit into other such existing frameworks-the paper may still be very valuable in that case.

We thank the referee for his/her comments. After our careful study of the three papers mentioned by the referee, we found that none of them are directly relevant to our statement "However, topology of the formerly reported ..." with $n\geq 3$ as indicated by the preceding sentences. The following are the descriptions of the references:

  • Phys. Rev. A 85, 042101 (2012) studies multiple EP2 rather than an EPn ($n\geq 3$) and thus is not related to the statement.
  • Phys. Rev. A 106, 012218 (2022) studies four-fold EPs, but these are merely accidental degeneracies rather than topological entities. As a result, they split into multiple EP2s in two dimensions.
  • Comm. Phys. 7, 109 (2024) computes $\pi$ Berry phase. But it is for EP2s rather than $n$-fold EP$n$s ($n\geq 3$).

As explained above in detail, these three references are not related to our statement "However, topology of ...". However, we agree that adding "$n\geq 3$" in the statement would be helpful to avoid misunderstanding. We also admit that Phys. Rev. A 106, 012218 (2022) discusses an EP4. Therefore, we have added $n\geq 3$ in the statement and cite the reference in the introductory sentences of EP$n$s ($n\geq 3$).

Reply to comment: 1-[3]

3- The notion of HEP would benefit from some important basic statements, especially around the introductory example (4).

Because Referee 1's comment on our model [Eq. (4)] is composed of multiple aspects, we separate it into four parts and address each of them one by one.

Reply to comment: 1-[3(a)]

-It appears that in the parameter space of eta1 and eta2, the degeneracy is a normal EP3s; if so this should be clearly stated. - Are these EPs also HEPs in the enlarged parameter space including the control parameters used to make the two HEPs meet and annihilate (such as phi and delta in he model of Eq (4))?

The referee asks two questions concerning the difference from the formerly reported EP$n$.

Regarding the referee's question "- It appears that in the parameter space [...] stated.", in the parameter space of $\zeta_1$ and $\zeta_2$, the degeneracy is identical to the formerly reported EP3 characterized by the resultant winding number. This is because non-Hermitian $n$-band touching is generically written as the $n\times n$ Jordan block form. The crucial difference form formerly reported EP$n$s is the Hopf topology of HEP$n$s which leads to the enhanced stability of HEP$n$ as discussed below. In response to the referee’s comment, we have added a footnote below Eq. (4) to clarify the relation/difference between the HEP3 and the formerly reported EP3.

Regarding the referee's question "- Are these EPs also HEPs [...] annihilate (such as phi and delta in he model of Eq (4))?", just introducing a parameter does not yield Hopf topology, which distinguishes the formerly reported EP$n$ and HEP$n$. These differences indicate the presence of multiple topological charges, leading to enhanced stability of HEP$n$ compared to formerly reported EP$n$. For instance, symmetry-protected HEP4 (or a loop of EP4 with the Hopf topology) remains stable in four dimensions, in contrast to a loop of formerly reported EP4 which can be contracted and annihilated. The above point is discussed in the previous version (see Sec. 5.1). However, we agree that further explanation would be helpful. Thus, we have added a footnote in Sec. 5.1. We also agree with the referee that the following point should be clearly stated: another choice of $\zeta$'s yields a formerly reported EP3 with winding topology. Thus, we have added a footnote below Eq. (4).

Reply to comment: 1-[3(b)]

-In the context it should be clarified how much of this is specific to EPs and how much is true for singularities of a suitable m-dimensional complex field. It seems much of this is tied to the mapping of the parameters to eta1 and eta2, and not particular to that we are dealing with an EP. Would any other physical property requiring eta1=eta2=0 do? The statement "The sought higher Hopf topology is then ensured by imposing the appropriate dependence [107] of the functions $\zeta_{1,2}(k)$ on momenta (parameters) k." seems to imply this, but I would prefer a more transparent statement, ideally already in the introduction. - For an analogously general perspective for a single (m=1) complex field see eg arXiv:2507.14373.

After carefully reading the referee's comment, we found that our previous formulation caused two confusions. We address them in the following.

Regarding the referee's comments "In the context it should be clarified how much [...]" and "It seems much of this is [...] ideally already in the introduction.", it is unclear what "eta1" and "eta2" in the referee's comment denote. If they denote matrix elements $\zeta_1$ and $\zeta_2$ of our model [Eq. (4)], our previous explanation seems to have caused confusion. Generically, the topology of EP$n$s is defined by the resultant vector rather than matrix elements themselves. Namely, what we require is the resultant vector possessing the nontrivial Hopf topology classified by $\pi_{c-1}(S^{m-1})$ away from HEPs ($m$ being the dimensions of the resultant vector and $c$ being codimension). The nontrivial topology of resultant vectors $\bf{R}$ on the sphere $S^{c-1}$ requires $\bf{R}=0$ at a point inside of the sphere which is nothing but an HEP. This fact leads to vanishing $\zeta$'s in our toy models (however, vanishing $\zeta$'s are not sufficient conditions for HEP formation for more general Hamiltonian forms). We also note that our toy model is prepared so that the resultants correspond to $\zeta$'s, which is explained in the previous version [see below Eq. (4)]. To clarify that the key ingredient is the resultant vector possessing Hopf topology, we have revised a sentence in the last paragraph of the introduction. If "eta1" and "eta2" in the referee's comment denote two generic complex parameters (i.e., four real parameters), the topology of these parameters is classified by $\pi_{4}(S^3)=\mathbb{Z}_2$ can be applied to phenomena other than EP$n$s. Indeed, mathematically similar arguments have been employed in the context of Hermitian topological insulators (see Refs. [114] and [115]). This fact is also mentioned in the original version [see below Eqs. (4), (8) and (13)].

Regarding the referee's comment "For an analogously general perspective for a single (m=1) complex field see eg arXiv:2507.14373", satisfying the constraint of complex value is already noted in terms of the discriminant Phys. Rev. Lett. 120, 146402 (2018) and Phys. Rev. Lett. 126, 086401 (2021). To clarify this point, we have added a footnote below Eq. (1).

Reply to comment: 1-[3(c)]

-In this context, the choice of the models should be clarified. Is their form dictated by the HEP use case, or do they have other features making them preferable over alternative models that may already exist in the literature?

We thank the referee for this constructive comment. As explained in the paragraph below Eq. (4), this model possesses a remarkable property: resultants are proportional to perturbations $r_j\propto \zeta_j$. This feature motivates us to analyze the model in Eq. (4) as the relation $r_j\propto \zeta_j$ leads to the clear correspondence between the momentum dependence of $\zeta$'s and Hopf topology of $\bf{R}$. However, we agree that a more detailed explanation is helpful. Thus, we have revised a sentence below Eq. (4).

Reply to comment: 1-[3(d)]

-Can a simple picture for the HEP annihilation been given? It may be something as simple as this: Consider an EP2 with Arnold-Jordan normal form (0,1;eps,0) and $\epsilon=k^2+\phi$, real k and phi. Then EPs annihilate at phi=0, k=0; there are two EPs at $\phi<0$ and none at $\phi>0$. While this is (hopefully!) not an HEP, a description of such form provides a much clearer picture, e.g. about the role of the underlying parameter space. Does something similar apply to HEPs?

It seems that the referee is confused EP2s with HEP$n$ $n\geq 3$. EP2s cannot acquire Hopf topology because their resultant vector is two-dimensional. This fact indicates that the referee's argument does not directly work. Pair annihilation of EP2s (2-fold band touching) with the same topological charge is a direct consequence of $\mathbb{Z}_2$ topology; a detailed analysis of the fusion of EP2s mentioned by the referee can be found in Phys. Rev. B 105, 085109 (2022). The novelty of HEP3s and symmetry-protected HEP5s is that the pair annihilation occurs for non-Hermitian $n$-fold band touching ($n=3,5$) with the same charge. In general, the occurrence of pair annihilation strongly supports the $\mathbb{Z}_2$ topology. We agree with the referee that this point should be stressed, although the novelty of this paper is the discovery of $n$-fold band touching ($n=3,5$) with $\mathbb{Z}_2$ topology. Thus, we have added as a corresponding footnote in the last paragraph of Sec. 2.

Reply to comment: 1-[4]

-4 "The specified perturbation of the Jordan form in the bottom row". The JNF is 0 in the bottom row; the authors use the Arnold Jordan normal form.

We thank the referee for pointing out our ambiguous explanation. After our careful check of the literature, we found the term "Arnold Jordan normal form" is not commonly used. Thus, we decided not to use "Arnold Jordan normal form". However, we agree that our previous explanation is also insufficient. Thus, to avoid ambiguity, we revised the corresponding sentence.

Reply to comment: 1-[5]

-5 The Arnold Jordan normal form, and possibly also the resultant approach, fail at EPs with geometric multiplicity $g>1$. Do HEPs with $g>1$ exist? For direct approach sidestepping the Arnold Jordan normal form and operating directly in parameter space of Hamiltonians see Physical Review Research 7 (2), 023062 (2025).

We thank the referee for his/her careful reading of our manuscript. Our resultant approach works regardless of the value of geometric multiplicity. Our resultant approach captures $n$-fold band-touching (i.e., algebraic multiplicity). Therefore, Hopf EPs are stable even when the geometric multiplicity is larger than $1$. Correspondingly, we do not need sidestepping as suggested by the referee. We also note that in general, geometric multiplicity becomes $1$. Namely, exceptional points with the geometric multiplicity larger than $1$ require fine-tuning of the matrix elements [see Phys. Rev. B 99, 121101(R)]. In order to clarify this point, we have added a footnote below Eq. (1).

Reply to comment: 1-[6]

-6 The reference list [76-84] introducing higher-order EPs as a concept should focus on original references such as W D Heiss 2008 J. Phys. A: Math. Theor. 41 244010, plus what would come up in a forward-backward literature search (including [83] is fine), instead of the self citations. The references also leave out the important insight into EPn arrived from sensing, see Photon. Res. 8, 1457 (2020) and reference forward and backwards from there.

We thank the referee for pointing out missing references. We have cited J. Phys. A: Math. Theor. 41 255206 (2008) as well as the references suggested by the referee [J. Phys. A: Math. Theor. 41 244010 (2008); Photon. Res. 8, 1457 (2020)]. In passing, we note that the term "multifold" is used to avoid potential confusion; the term "higher-order" is used in another sense (i.e, higher-order topological insulator).

The referee also commented on citations of [76-82] and [84]. Although the multifold exceptional points were analyzed more than ten years ago, their topological protection is revealed by these recent works. We appreciate the referee's comment pointing out our ambiguous explanations. In response to this referee's comment, we have revised the sentence in the third paragraph of the introduction.

Reply to comment: 1-[7]

-7 "Reference 78 has pointed out that the codimension of symmetry-protected EP4s is three,..." It is known for a much longer time that EPns have codimension 2n-2 (it follows directly from the Arnold Jordan Normal form).

Although the referee wrote that the codimension $2n-2$ is known for a much longer time (without providing any reference), his/her statement does not reproduce the correct result: for symmetry-protected EP$n$s with $n=4$, the formula suggested by the referee predicts codimension $2n-2=6$ instead of the correct value $3$. For this reason, we decided to maintain our original statement.

Reply to comment: 1-[8]

-8 Generally, a more thorough approach to referencing in this field is required. This is not so much as expanding the list in this active field, but giving proper credit to the original works establishing key knowledge.

We thank the referee for bringing our attention to missing references. In addition to the references mentioned by Referee 1, we have cited a relevant reference [W.D. Heiss, Phys. Rev. E 58, 2894 (1998)].

-

In closing, we would like to thank Referee 1 again for their careful review of our manuscript and invaluable comments. We believe that we have appropriately addressed all the comments by Referee 1. We do hope that the above reply and the corresponding revisions of our manuscript will meet with the approval of Referee 1 for publication of our manuscript in SciPost Physics.

Attachment:

diff_SciPost20251029.pdf

Author:  Tsuneya Yoshida  on 2025-10-26  [id 5950]

(in reply to Report 1 on 2025-08-04)

Warnings issued while processing user-supplied markup:

  • Inconsistency: Markdown and reStructuredText syntaxes are mixed. Markdown will be used.
    Add "#coerce:reST" or "#coerce:plain" as the first line of your text to force reStructuredText or no markup.
    You may also contact the helpdesk if the formatting is incorrect and you are unable to edit your text.

Reply to Referee 1's comments

This paper describes topological properties of a certain class of exceptional points embedded in a suitable parameter space. It is a timely contribution to a topical field, with the main drawback being the lack of connections to an ample body of existing literature. This also impacts on the assessment of its novelty, which should be reserved until the clarifications listed below are made.

We would like to thank Referee 1 for their careful reading of our manuscript. We also appreciate their comments which were helpful in improving our manuscript. After carefully studying all the comments by the referee, we found that our insufficient explanations in the previous version prevented them from assessing the novelty. The novelty of our results is discovering a novel type of $n$-fold EPs (EP$n$) with $n\geq 3$ dubbed Hopf EP$n$s. In particular, Hopf EP3s and symmetry-protected HEP5s possess $\mathbb{Z}_2$ topology in contrast to formally reported EP$n$s. This point is supported by Referee B, stating "This article introduces a new class of exceptional points (EP), dubbed Hopf exceptional points. While previous topological classifications of exceptional points have, to my knowledge, only involved integer invariants Z, the exceptional points identified in this manuscript are instead characterized by finite groups such as Z2 and Z3, involving a very different robustness to continuous perturbations that may or may not annihilate them." and "I find this work highly pertinent and timely. It represents a very interesting extension of the existing topological classification of exceptional points. I therefore recommend it for publication." To make the novelty clear, we have substantially revised the manuscript by taking all the comments into account. In the following, we reply to them one by one.

Reply to comment: 1-[1]

1- The introduction and later parts of the paper mixes different topological notions - quasiparticles from fermionic Fock space topology and singularities of real and complex fields, such as diabolic points and exceptional points. See also phrases such as "meaning that they act as their own antiparticle...", "Notably, the HEP acts as its own antiparticle...". A quasiparticle is a very well-defined concept; it has a mass, can move, etc, and all of this does not apply here. On the other hand, static singularities of complex fields [i the mathematical sense] have well defined properties, too, and these apply very well to EPs, also in the variant described here. I understand the motivation being the annihilation property of the HEPs. However, a clear distinction of these only superficially overlapping concepts would add clarity. This also impacts on the description and understanding of the nature of HEPs and choice of models; see further below.

We thank the referee for pointing out the potential confusion between quasiparticles in the traditional condensed matter sense (quasiparticles with mass and mobility, etc.) and band touching (singularity of the fields). In our manuscript, the term "quasiparticle" is used in the broader sense common in topological condensed matter physics where it denotes topologically protected excitations or band singularities, irrespective of whether they have a conventional particle-like dispersion. This broader definition is widely used in the literature, e.g., for Weyl points and Majorana zero modes [see references Rev. Mod. Phys. {\bf 82}, 3045 (2010); Rev. Mod. Phys. {\bf 83}, 1057 (2011); Science {\bf 353}, 6299 (2016)]. Such topologically protected "quasiparticles" include HEP$n$s as well as exceptional points. To avoid ambiguity, we add a footnote in the first paragraph of Sec. 1 explaining that "quasiparticle" in our manuscript refers to a topological quasiparticle which means a topologically protected excitation or band touching.

Reply to comment: 1-[2]

2- "However, topology of the formerly reported n-fold exceptional points (EPn’s) is generally characterized by Z invariants [77, 78]." This should make connection to existing EP topology as developed, eg, by the JW Ryu group, Phys. Rev. A 85, 042101 (2012); Phys. Rev. A 106, 012218 (2022); Commun. Phys. 7, 109 (2024) [possibly also other groups]. For instance, the last paper states in the abstract "our proposed classification gives rise to finer Z2 classifications depending on the presence of a $\pi$ Berry phase after the encircling of the EPs." Please clarify, in detail, how this relates to the HEPs described here. The authors should not shy away to clarify if their results happen to perfectly fit into other such existing frameworks-the paper may still be very valuable in that case.

We thank the referee for his/her comments. After our careful study of the three papers mentioned by the referee, we found that none of them are directly relevant to our statement "However, topology of the formerly reported ..." with $n\geq 3$ as indicated by the preceding sentences. The following are the descriptions of the references: - * Phys. Rev. A {\bf 85}, 042101 (2012) studies multiple EP2 rather than an EPn ($n\geq 3$) and thus is not related to the statement. * Phys. Rev. A {\bf 106}, 012218 (2022) studies four-fold EPs, but these are merely accidental degeneracies rather than topological entities. As a result, they split into multiple EP2s in two dimensions. * Comm. Phys. {\bf 7}, 109 (2024) computes $\pi$ Berry phase. But it is for EP2s rather than $n$-fold EP$n$s ($n\geq 3$). - As explained above in detail, these three references are not related to our statement "However, topology of ...". However, we agree that adding "$n\geq 3$" in the statement would be helpful to avoid misunderstanding. We also admit that Phys. Rev. A {\bf 106}, 012218 (2022) discusses an EP4. Therefore, we have added $n\geq 3$ in the statement and cite the reference in the introductory sentences of EP$n$s ($n\geq 3$).

Reply to comment: 1-[3]

3- The notion of HEP would benefit from some important basic statements, especially around the introductory example (4).

Because Referee's comment 3 on our model [Eq. (4)] is composed of multiple aspects, we separate it into four parts and address each of them one by one.

Reply to comment: 1-[3(a)]

-It appears that in the parameter space of eta1 and eta2, the degeneracy is a normal EP3s; if so this should be clearly stated. - Are these EPs also HEPs in the enlarged parameter space including the control parameters used to make the two HEPs meet and annihilate (such as phi and delta in he model of Eq (4))?

The referee asks two questions concerning the difference from the formerly reported EP$n$.

Regarding the referee's question "- It appears that in the parameter space [...] stated.", in the parameter space of $\zeta_1$ and $\zeta_2$, the degeneracy is identical to the formerly reported EP3 characterized by the resultant winding number. This is because non-Hermitian $n$-band touching is generically written as the $n\times n$ Jordan block form. The crucial difference form formerly reported EP$n$s is that Hopf topology of HEP$n$s which leads to the enhanced stability of HEP$n$ as discussed below. In response to the referee’s comment, we have added a footnote below Eq. (4) to clarify the relation/difference between the HEP3 and the formerly reported EP3.

Regarding the referee's question "- Are these EPs also HEPs [...] annihilate (such as phi and delta in he model of Eq (4))?", just introducing a parameter does not yield Hopf topology, which distinguishes the formerly reported EP$n$ and HEP$n$. These differences indicate the presence of multiple topological charges, leading to enhanced stability of HEP$n$ compared to formerly reported EP$n$. For instance, symmetry-protected HEP4 (or a loop of EP4 with the Hopf topology) remains stable in four dimensions, in contrast to a loop of formerly reported EP4 which can be contracted and annihilated. The above point is discussed in the previous version (see Sec. 5.1). However, we agree that further explanation would be helpful. Thus, we have added a footnote in Sec. 5.1. We also agree with the referee that the following point should be clearly stated: another choice of $\zeta$'s yields a formerly reported EP3 with winding topology. Thus, we have added a footnote below Eq. (4).

Reply to comment: 1-[3(b)]

-In the context it should be clarified how much of this is specific to EPs and how much is true for singularities of a suitable m-dimensional complex field. It seems much of this is tied to the mapping of the parameters to eta1 and eta2, and not particular to that we are dealing with an EP. Would any other physical property requiring eta1=eta2=0 do? The statement "The sought higher Hopf topology is then ensured by imposing the appropriate dependence [107] of the functions $\zeta_{1,2}(k)$ on momenta (parameters) k." seems to imply this, but I would prefer a more transparent statement, ideally already in the introduction. - For an analogously general perspective for a single (m=1) complex field see eg arXiv:2507.14373.

After carefully reading the referee's comment, we found that our previous formulation caused two confusions. We address them in the following.

Regarding the referee's comments "In the context it should be clarified how much [...]" and "It seems much of this is [...] ideally already in the introduction.", it is unclear what "eta1" and "eta2" in the referee's comment denote. If they denote matrix elements $\zeta_1$ and $\zeta_2$ of our model [Eq. (4)], our previous explanation seems to have caused confusion. Generically, the topology of EP$n$s is defined by the resultant vector rather than matrix elements themselves. Namely, what we require is the resultant vector possessing the nontrivial Hopf topology classified by $\pi_{c-1}(S^{m-1})$ away from HEPs ($m$ being the dimensions of the resultant vector and $c$ being codimension). The nontrivial topology of resultant vectors $\bf{R}$ on the sphere $S^{c-1}$ requires $\bf{R}=0$ at a point inside of the sphere which is nothing but an HEP. This fact leads to vanishing $\zeta$'s in our toy models (however, vanishing $\zeta$'s are not sufficient conditions for HEP formation for more general Hamiltonian forms). We also note that our toy model is prepared so that the resultants correspond to $\zeta$'s, which is explained in the previous version [see below Eq. (4)]. To clarify that the key ingredient is resultant vectors possessing Hopf topology, we have revised a sentence in the last paragraph of the introduction. If "eta1" and "eta2" in the referee's comment denote two generic complex parameters (i.e., four real parameters), the topology of these parameters is classified by $\pi_{4}(S^3)=\mathbb{Z}_2$ can be applied to phenomena other than EP$n$s. Indeed, mathematically similar arguments have been employed in the context of Hermitian topological insulators (see Refs. [114] and [115]). This fact is also mentioned in the original version [see below Eqs. (4), (8) and (13)].

Regarding the referee's comment "For an analogously general perspective for a single (m=1) complex field see eg arXiv:2507.14373", satisfying the constraint of complex value is already noted in terms of the discriminant Phys. Rev. Lett. {\bf 120}, 146402 (2018) and Phys. Rev. Lett. {\bf 126}, 086401 (2021). To clarify this point, we have added a footnote below Eq. (1).

Reply to comment: 1-[3(c)]

-In this context, the choice of the models should be clarified. Is their form dictated by the HEP use case, or do they have other features making them preferable over alternative models that may already exist in the literature?

We thank the referee for this constructive comment. As explained in the paragraph below Eq. (4), this model possesses a remarkable property: resultants are proportional to perturbations $r_j\propto \zeta_j$. This feature motivates us to analyze the model in Eq. (4) as the relation $r_j\propto \zeta_j$ leads to the clear correspondence between the momentum dependence of $\zeta$'s and Hopf topology of $\bf{R}$. However, we agree that a more detailed explanation is helpful. Thus, we have revised a sentence below Eq. (4).

Reply to comment: 1-[3(d)]

-Can a simple picture for the HEP annihilation been given? It may be something as simple as this: Consider an EP2 with Arnold-Jordan normal form (0,1;eps,0) and $\epsilon=k^2+\phi$, real k and phi. Then EPs annihilate at phi=0, k=0; there are two EPs at $\phi<0$ and none at $\phi>0$. While this is (hopefully!) not an HEP, a description of such form provides a much clearer picture, e.g. about the role of the underlying parameter space. Does something similar apply to HEPs?

It seems that the referee is confused EP2s with HEP$n$ $n\geq 3$. EP2s cannot acquire Hopf topology because their resultant vector is two-dimensional. This fact indicates that the referee's argument does not directly work. Pair annihilation of EP2s (2-fold band touching) with the same topological charge is a direct consequence of $\mathbb{Z}_2$ topology; a detailed analysis of the fusion of EP2s mentioned by the referee can be found in Phys. Rev. B {\bf 105}, 085109 (2022). The novelty of HEP3s and symmetry-protected HEP5s is that the pair annihilation occurs for non-Hermitian $n$-fold band touching ($n=3,5$) with the same charge. In general, the occurrence of pair annihilation strongly supports the $\mathbb{Z}_2$ topology. We agree with the referee that this point should be stressed, although the novelty of this paper is the discovery of $n$-fold band touching ($n=3,5$) with $\mathbb{Z}_2$ topology. Thus, we have added as a corresponding footnote in the last paragraph of Sec. 2.

Reply to comment: 1-[4]

-4 "The specified perturbation of the Jordan form in the bottom row". The JNF is 0 in the bottom row; the authors use the Arnold Jordan normal form.

We thank the referee for pointing out our ambiguous explanation. After our careful check of the literature, we found the term "Arnold Jordan normal form" is not commonly used. Thus, we decided not to use "Arnold Jordan normal form". However, we agree that our previous explanation is also insufficient. Thus, to avoid ambiguity, we revised the corresponding sentence.

Reply to comment: 1-[5]

-5 The Arnold Jordan normal form, and possibly also the resultant approach, fail at EPs with geometric multiplicity $g>1$. Do HEPs with $g>1$ exist? For direct approach sidestepping the Arnold Jordan normal form and operating directly in parameter space of Hamiltonians see Physical Review Research 7 (2), 023062 (2025).

We thank the referee for his/her careful reading of our manuscript. Our resultant approach works regardless of the value of geometric multiplicity. Our resultant approach captures $n$-fold band-touching (i.e., algebraic multiplicity). Therefore, Hopf EPs are stable even when the geometric multiplicity is larger than $1$. Correspondingly, we do not need sidestepping as suggested by the referee. We also note that in general, geometric multiplicity becomes $1$. Namely, exceptional points with the geometric multiplicity larger than $1$ require fine-tuning of the matrix elements [see Phys. Rev. B {\bf 99}, 121101(R)]. In order to clarify this point, we have added a footnote below Eq. (1).

Reply to comment: 1-[6]

-6 The reference list [76-84] introducing higher-order EPs as a concept should focus on original references such as W D Heiss 2008 J. Phys. A: Math. Theor. 41 244010, plus what would come up in a forward-backward literature search (including [83] is fine), instead of the self citations. The references also leave out the important insight into EPn arrived from sensing, see Photon. Res. 8, 1457 (2020) and reference forward and backwards from there.

We thank the referee for pointing out missing references. We have cited J. Phys. A: Math. Theor. {\bf 41} 255206 (2008) as well as the references suggested by the referee [J. Phys. A: Math. Theor. {\bf 41} 244010 (2008); Photon. Res. {\bf 8}, 1457 (2020)]. In passing, we note that the term "multifold" is used to avoid potential confusion; the term "higher-order" is used in another sense (i.e, higher-order topological insulator).

The referee also commented on citations of [76-82] and [84]. Although the multifold exceptional points were analyzed more than ten years ago, their topological protection is revealed by these recent works. We appreciate the referee's comment pointing out our ambiguous explanations. In response to this referee's comment, we have revised the sentence in the third paragraph of the introduction.

Reply to comment: 1-[7]

-7 "Reference 78 has pointed out that the codimension of symmetry-protected EP4s is three,..." It is known for a much longer time that EPns have codimension 2n-2 (it follows directly from the Arnold Jordan Normal form).

Although the referee wrote that the codimension $2n-2$ is known for a much longer time (without providing any reference), his/her statement does not reproduce the correct result: for symmetry-protected EP$n$s with $n=4$, the formula suggested by the referee predicts codimension $2n-2=6$ instead of the correct value $3$. For this reason, we decided to maintain our original statement.

Reply to comment: 1-[8]

-8 Generally, a more thorough approach to referencing in this field is required. This is not so much as expanding the list in this active field, but giving proper credit to the original works establishing key knowledge.

We thank the referee for bringing our attention to missing references. We have studied all of the references mentioned by the referee and cited the relevant one [W.D. Heiss, Phys. Rev. E {\bf 58}, 2894 (1998)].

-

In closing, we would like to thank Referee 1 again for their careful review of our manuscripts and invaluable comments. We believe that we have appropriately addressed all the comments by Referee 1. We do hope that the above reply and the corresponding revisions of our manuscript will meet with the approval of Referee 1 for publication of our manuscript in SciPost Physics.

Reply to Referee 2's comments

This article introduces a new class of exceptional points (EP), dubbed Hopf exceptional points. While previous topological classifications of exceptional points have, to my knowledge, only involved integer invariants Z, the exceptional points identified in this manuscript are instead characterized by finite groups such as Z2 and Z3, involving a very different robustness to continuous perturbations that may or may not annihilate them. The possibility of these new classes of exceptional points arises from considering multiple exceptional points in dimension higher than two, for systems possibly constrained by anti-unitary symmetries. Their topological classification is then determined by the winding properties of a “resultant vector.” This vector had previously been introduced in particular to characterize higher-order and symmetry-protected exceptional points, by considering homotopy groups of the form

$$ S^{D-1} \to S^{D-1}, $$
where $D$ is the dimension of the system in which the exceptional point appears as an isolated defect. Here, however, different homotopy groups of spheres are employed, which allows one to understand the stabilization of exceptional points in higher dimensions. For instance, in four dimensions and in the presence of an anti-unitary symmetry, the earlier classification accounts only for the existence of lines of EP4, while the classification proposed in this article identifies isolated EP4s. These new exceptional points are termed Hopf exceptional points because the mappings defined here by the resultant vector are classified by the groups
$$ \pi_4(S^3) \ \ \text{and} \ \ \pi_3(S^2) $$
whose associated homotopy invariant is the Hopf index.

We thank the referee for evaluating the significance of our manuscript. As the referee noted, the novelty of our work lies in the discovery of $n$-fold exceptional points with Hopf topology which we term Hopf exceptional points. In contrast to the previous works, Hopf exceptional points can exhibit a variety of topological structures (e.g., $\mathbb{Z}_2$, $\mathbb{Z}_3$, or $\mathbb{Z}_{12}$) even at generic points in momentum space, which gives rise to exotic annihilation properties.

Reply to comment: 2-[1]

My only concern is about the validity of using the Hopf indices (3) and (12) as relevant topological invariants for any homotopy group of the spheres, as it is implicitly suggested in the end of the manuscript. The Hopf fibration being given by the map from $S^{2n-1}\to P(C^n)$ (the projective space of $C^n$), I understand that e.g. when $n=2$, one gets the map $S^3 \to S^2$ characterized by the homotopy group $\pi_3(S^2)$ as that used in this manuscript, but I doubt that all the other cases mentioned in the end of this article are Hopf maps. Could they authors justify the homotopy invariant for any homotopy group of the spheres? Is that a Hopf index? What is the expression of a Hopf index beyond the maps from $S^3 \to S^2$ and $S^4 \to S^3$? It seems to me that the authors have identified a way to characterize as many classes of EPs as homotopy groups of spheres, and only very few are Hopf.

We thank the referee for pointing out our insufficient explanation.

First, we note that the topological invariant in Eq. (3) [Eq. (12)] is applicable to HEP$n$s whose resultant vector possesses nontrivial topology of the map $S^3\to S^2$ [$S^5\to S^4$]. Topological invariants for the other cases of maps in Table I remain an open question as noted in the last paragraph in the original version.

We used the term "Hopf" to denote the standard Hopf map and its generalization (i.e., topologically nontrivial maps between higher-dimensional spheres). For compactness of terminology, we collectively refer to this family of exceptional points as Hopf exceptional points.

We agree with the referee that the above points should be clearly explained. In response to the referee's comment, we have added footnote 2 and a sentence at the end of Sec. 5.3.

Reply to comment: 2-[2]

A part from this point, I find this work highly pertinent and timely. It represents a very interesting extension of the existing topological classification of exceptional points. I therefore recommend it for publication.

We again thank Referee B for reviewing our manuscript. We are happy to hear that the referee recommends the publication in the journal SciPost Physics.

---

## Round 1 · Referee Report · Anonymous (Referee 2) · 2025-9-18

Strengths

1- extends in an original way the topological classification of exceptional points 2- gives stimulating directions for further investigations

Weaknesses

1- Lack of physical models, i.e. beyond toy models, to illustrate the main results.

Report

This article introduces a new class of exceptional points (EP), dubbed *Hopf exceptional points*. While previous topological classifications of exceptional points have, to my knowledge, only involved integer invariants $Z$, the exceptional points identified in this manuscript are instead characterized by finite groups such as $Z_2$ and $Z_3$, involving a very different robustness to continuous perturbations that may or may not annihilate them. The possibility of these new classes of exceptional points arises from considering multiple exceptional points in dimension higher than two, for systems possibly constrained by anti-unitary symmetries. Their topological classification is then determined by the winding properties of a “resultant vector.” This vector had previously been introduced in particular to characterize higher-order and symmetry-protected exceptional points, by considering homotopy groups of the form
$$ S^{D-1} \to S^{D-1}, $$
where $D$ is the dimension of the system in which the exceptional point appears as an isolated defect.

Here, however, different homotopy groups of spheres are employed, which allows one to understand the stabilization of exceptional points in higher dimensions. For instance, in four dimensions and in the presence of an anti-unitary symmetry, the earlier classification accounts only for the existence of lines of EP4, while the classification proposed in this article identifies isolated EP4s. These new exceptional points are termed *Hopf exceptional points* because the mappings defined here by the resultant vector are classified by the groups

$$ \pi_4(S^3) \quad \text{and} \quad \pi_3(S^2), $$

whose associated homotopy invariant is the Hopf index.

My only concern is about the validity of using the Hopf indices (3) and (12) as relevant topological invariants for any homotopy group of the spheres, as it is implicitly suggested in the end of the manuscript. The Hopf fibration being given by the map from $S^{2n-1} \to P(C^n)$ (the projective space of $C^n$), I understand that e.g. when n=2, one gets the map $S^3 \to S^2$ characterized by the homotopy group $\pi_3(S^2)$ as that used in this manuscript, but I doubt that all the other cases mentioned in the end of this article are Hopf maps. Could they authors justify the homotopy invariant for any homotopy group of the spheres? Is that a Hopf index? What is the expression of a Hopf index beyond the maps from $S^3\to S^2$ and $S^4 \to S^3$ ? It seems to me that the authors have identified a way to characterize as many classes of EPs as homotopy groups of spheres, and only very few are Hopf.

A part from this point, I find this work highly pertinent and timely. It represents a very interesting extension of the existing topological classification of exceptional points. I therefore recommend it for publication.

Recommendation

Ask for minor revision

  • validity: high
  • significance: high
  • originality: high
  • clarity: high
  • formatting: good
  • grammar: perfect

Author:  Tsuneya Yoshida  on 2025-10-29  [id 5967]

(in reply to Report 2 on 2025-09-18)

Reply to Referee 2's comments

This article introduces a new class of exceptional points (EP), dubbed Hopf exceptional points. While previous topological classifications of exceptional points have, to my knowledge, only involved integer invariants Z, the exceptional points identified in this manuscript are instead characterized by finite groups such as Z2 and Z3, involving a very different robustness to continuous perturbations that may or may not annihilate them. The possibility of these new classes of exceptional points arises from considering multiple exceptional points in dimension higher than two, for systems possibly constrained by anti-unitary symmetries. Their topological classification is then determined by the winding properties of a “resultant vector.” This vector had previously been introduced in particular to characterize higher-order and symmetry-protected exceptional points, by considering homotopy groups of the form

$$ S^{D-1} \to S^{D-1}, $$
where $D$ is the dimension of the system in which the exceptional point appears as an isolated defect. Here, however, different homotopy groups of spheres are employed, which allows one to understand the stabilization of exceptional points in higher dimensions. For instance, in four dimensions and in the presence of an anti-unitary symmetry, the earlier classification accounts only for the existence of lines of EP4, while the classification proposed in this article identifies isolated EP4s. These new exceptional points are termed Hopf exceptional points because the mappings defined here by the resultant vector are classified by the groups
$$ \pi_4(S^3) \ \ \text{and} \ \ \pi_3(S^2) $$
whose associated homotopy invariant is the Hopf index.

We thank the referee for evaluating the significance of our manuscript. As the referee noted, the novelty of our work lies in the discovery of $n$-fold exceptional points with Hopf topology which we term Hopf exceptional points. In contrast to the previous works, Hopf exceptional points (HEP$n$ with $n\geq 2$) can exhibit a variety of topological structures (e.g., $\mathbb{Z}_2$, $\mathbb{Z}_3$, or $\mathbb{Z}_{12}$) even at generic points in momentum space, which gives rise to exotic annihilation properties.

We also thank the referees for pointing out our insufficient explanation. Taking into account all of the comments by both referees, we have revised the manuscript. We also attach our revised manuscript with the changes highlighted.

In the following, we address the revisions corresponding to the comments by Referee 2.

Reply to comment: 2-[1]

My only concern is about the validity of using the Hopf indices (3) and (12) as relevant topological invariants for any homotopy group of the spheres, as it is implicitly suggested in the end of the manuscript. The Hopf fibration being given by the map from $S^{2n-1}\to P(C^n)$ (the projective space of $C^n$), I understand that e.g. when $n=2$, one gets the map $S^3 \to S^2$ characterized by the homotopy group $\pi_3(S^2)$ as that used in this manuscript, but I doubt that all the other cases mentioned in the end of this article are Hopf maps. Could they authors justify the homotopy invariant for any homotopy group of the spheres? Is that a Hopf index? What is the expression of a Hopf index beyond the maps from $S^3 \to S^2$ and $S^4 \to S^3$? It seems to me that the authors have identified a way to characterize as many classes of EPs as homotopy groups of spheres, and only very few are Hopf.

We thank the referee for pointing out our insufficient explanation.

First, we note that the topological invariant in Eq. (3) [Eq. (12)] is applicable to HEP$n$s whose resultant vector possesses nontrivial topology of the map $S^3\to S^2$ [$S^5\to S^4$]. Topological invariants for the other cases of maps in Table I remain an open question as noted in the last paragraph in the original version.

We used the term "Hopf" to denote the standard Hopf map and its generalization (i.e., topologically nontrivial maps between higher-dimensional spheres). For compactness of terminology, we collectively refer to this family of exceptional points as Hopf exceptional points.

We agree with the referee that the above points should be clearly explained. In response to the referee's comment, we have added footnote 2 and a sentence at the end of Sec. 5.3.

Reply to comment: 2-[2]

A part from this point, I find this work highly pertinent and timely. It represents a very interesting extension of the existing topological classification of exceptional points. I therefore recommend it for publication.

We again thank Referee 2 for reviewing our manuscript. We are happy to hear that the referee recommends the publication in the journal SciPost Physics.

---

## Editorial Decision

resubmitted